# The autumnal lockdown was not the main initiator of the decrease in SARS-CoV-2 circulation in France

Veronica Pereda-Loth [1,2], Aldair Martínez Pineda[1], Lenka Tisseyre[1,2], Monique Courtade-Saidi [2], Christophe Bousquet [3], Camille Ferdenzi[3], Thierry Letellier[1], Moustafa Bensafi [3] & Denis Pierron [1✉]

### Abstract

**Background** In response to the SARS-CoV-2 pandemic, governments have taken drastically restrictive public health measures with significant collateral effects. It is important to understand the impact of these measures on SARS-CoV-2 circulation. However, pandemic indicators lag behind the actual level of viral circulation and these delays are an obstacle to assessing the effectiveness of policy decisions. Here, we propose one way to solve this problem by synchronizing the indicators with viral circulation in a country (France) based on a landmark event.

**Methods** Based on a first lockdown, we measured the time lag between the peak of governmental and non-governmental surveillance indicators and the highest level of virus circulation. This allowed alignment of all surveillance indicators with viral circulation during the second period of the epidemic, overlaid with the type of public health measures implemented.

**Results** We show that the second peak in viral circulation in France happened ~21 October 2020, during the public health state of emergency but before the lockdown (31 October). Indicators also suggest that viral circulation decreased earlier in locations where curfews were implemented. Indicators did, however, begin to rise once the autumnal lockdown was lifted and the state of emergency resumed.

**Conclusions** Overall, these results suggest that in France, the 2020 autumnal lockdown was not the main initiator of the decrease in SARS-CoV-2 circulation and curfews were important in achieving control of the transmission. Less-restrictive measures may need to be balanced with more-stringent measures to achieve desirable public health outcomes over time.

### Plain Language Summary

Non-pharmaceutical measures to control the SARS-CoV-2 pandemic and prevent the saturation of health-care systems have a negative impact on physical and mental health as well as on the economy. Therefore, it is important to better understand how each measure influences the circulation of the virus. In this paper, we analyze governmental and non-governmental data and show that the level of circulation of SARS-CoV-2 in France decreased in autumn 2020 after the implementation of the state of emergency measures and curfews but before the implementation of the lockdown. We also show that locations where a curfew was implemented experienced an earlier decrease in viral circulation. However, circulation of the virus has increased following release of the lockdown and return to state of emergency measures. This shows that less stringent measures may be sufficient to trigger a decrease in viral circulation, but more restrictive measures may be needed to maintain suppression over time.

[1] Équipe de Médecine Evolutive, URU EVOLSAN, Faculté de Chirurgie Dentaire, Université Toulouse III, Toulouse, France. [2] GSBMS, faculté de médecine Rangueil, Université Toulouse III, Toulouse, France. [3] Lyon Neuroscience Research Center, CNRS UMR5292, INSERM U1028, Université Claude Bernard Lyon 1, Bron, France. ✉email: denis.pierron@univ-tlse3.fr

During the past year, SARS-CoV-2 has spread across the world, resulting in a tremendous loss of human lives and overburdened healthcare systems[1]. In response, many governments have taken measures (state of emergency, curfew, and lockdown) drastically restricting the freedom of their citizens to prevent overloading hospitals. The measures with the highest degree of restriction have the heaviest societal and economic costs and also have a negative impact on general health and well-being[2]. For instance, lockdowns cost more than an overnight curfew or a state of emergency[3]. Therefore, in order to minimize these collateral effects on health and society, it is of vital importance to better understand the impact of these measures on the circulation of SARS-CoV-2. From October to December 2020, France gradually implemented measures starting with a state of emergency, which is the least restrictive, followed by a curfew and finally a lockdown (see Supplementary Data 1). This provided the opportunity to verify whether virus diffusion was altered after the beginning of the most restrictive measures (lockdown), or if it had already been checked by less-restrictive measures such as the state of emergency and/or the curfew.

To track the changes in SARS-CoV-2 viral circulation in France, several epidemic surveillance indicators are currently used. These include the ratio of consultations for suspected cases of COVID-19 to general consultations at the emergency room (ER), the number of hospitalizations, the number of admissions to critical care resuscitation units (CCRU), and the number of deaths per day caused by COVID-19. However, all of these indicators lag behind the actual level of viral circulation. For example, several weeks elapse between the death of an individual and the initial viral infection[4]. These delays are an obstacle to assessing the level of viral circulation in a given territory on a daily basis and to relating it to specific events or policy-making decisions.

One way to solve this problem is to synchronize these indicators with viral circulation based on a landmark event. The case of the French population provides a useful framework to address this issue, indeed in 2020 two periods of heavy SARS-CoV-2 circulation have been reported based on the number of deaths and cases (Supplementary Figure 1). In spring, the epidemic was characterized in France by a sudden increase and decrease in viral circulation. The first lockdown prevented the virus from circulation at its highest level on 17th March[1,5–7]. Based on this event, we measured the time lag between the peak of each surveillance indicator and the highest level of virus circulation. Hence, we were able to synchronize and timely phase all surveillance indicators with the actual level of virus circulation. This allowed us to monitor the circulation of the SARS-CoV-2 virus during the second period of the epidemic in France and overlay the governmental response (from a state of emergency to curfew and then to lockdown). We show that the second peak in viral circulation in France happened ~21 October 2020, during the public health state of emergency but before the autumnal lockdown. Indicators also suggest that viral circulation decreased earlier in locations where curfews were implemented and began to rise once the autumnal lockdown was lifted.

## Methods
**Data set**. Governmental indicators regarding the healthcare system in France (ER consultation, hospitalizations, CCRU admission and deaths, PCR) were downloaded on 25 February 2021 from the French Public Health website https://www.data.gouv.fr/fr/pages/donnees-coronavirus. Hospitalization, CCRU admission, and death indicators correspond to the new number of cases by day. ER consultation ratio corresponds to the ratio of consultation with a medical diagnosis of COVID-19 over the general consultations at the ER in hospitals. A number of detected new viral infections (positive and negative) based on nucleic acid are also downloaded from the French Public Health website. We report the raw number as well as the rate of a daily positive diagnosis.

An online survey was conducted in the French population between 8 April and 10 January 2021 and aimed at characterizing chemosensory disorders in people with and without COVID-19, as well as their consequences on quality of life. In all, 4628 responses were analyzed (Supplementary Data 2). This survey was approved by the CNRS ethics committee. Data collection was strictly anonymous. The protocol complies with the revised Declaration of Helsinki and the study was approved by the ethics committee of the Institute of Biological Sciences of the CNRS on the 3rd of April 2020 (DPO #TRRECH-467). All individuals provided informed consent when participating in the survey.

The study of online queries was downloaded from Google Trends (https://trends.google.com/), using the R library gtrendsR. We looked separately for the popularity of the terms: loss of smell (perte odorat) and loss of taste (perte goût), using default selection of *All categories* and *Web search*), within the timeframe of 1 March 2020 to 10 January 2021. Google Trends do not provide the actual numbers of searches but rather a relative score from 0 to 100 (100 correspondings to the day with the greatest number of searches during the specified time period).

**Statistics**. Phasing study: data were analyzed using R software (4.0) and its standard packages (maps, ggplot, etc.). In order to account for weekend reporting effects and random fluctuation, we computed a rolling average over 7 days, for each indicator. Each indicator displaying its annual maximum during the spring lockdown, we computed the number of days between the day before the start of this first lockdown (17th march) and the annual maximum of the indicator (its peak). We considered the number of days computed as the delay between the peak of the indicator and the peak of circulation of the virus. Then, we phased all indicators (i.e., 1–google loss of smell; 2–google loss of taste; 3–report of change in smell; ER consultation, hospitalization, CCRU admission, COVID deaths) to represent virus circulation on specific days (Fig. 1). We also define a composite indicator by computing the mean of all indicators and compute its rolling slope on 7 days. All indicators were weighted equally. Virologic tests based on PCR were not available nationally during the first lockdown; we did not include them in this first analysis.

Local study: pandemic indicator data at the departmental level were downloaded from the French public health database (Supplementary Data 2). For each indicator, we computed a rolling average over 7 days at the department level. For each indicator, we report the date of the maximal value from 1 October to 28 November. Curfew dates by the department were found on the official government website https://www.gouvernement.fr/info-coronavirus. Curfew was declared on the whole Paris region (Ile-de-France) and major metropolises and later on specific departments. Metropolises represent a cluster of cities merged together, for example, Toulouse metropolis regroups 37 cities and 750 000 inhabitants. Metropolises and departments are two distinct french administrative entities since a department includes metropolises as well as less populated countryside regions. Epidemiologic data being available at the department scale, we assimilated metropolises and department. This is a reasonable extension since the metropolises not only regroup most of the inhabitants, it also regroups most of the night activity as well as the population impacted by curfew such as students. Moreover, considering the results (importance of the early curfew), this is a

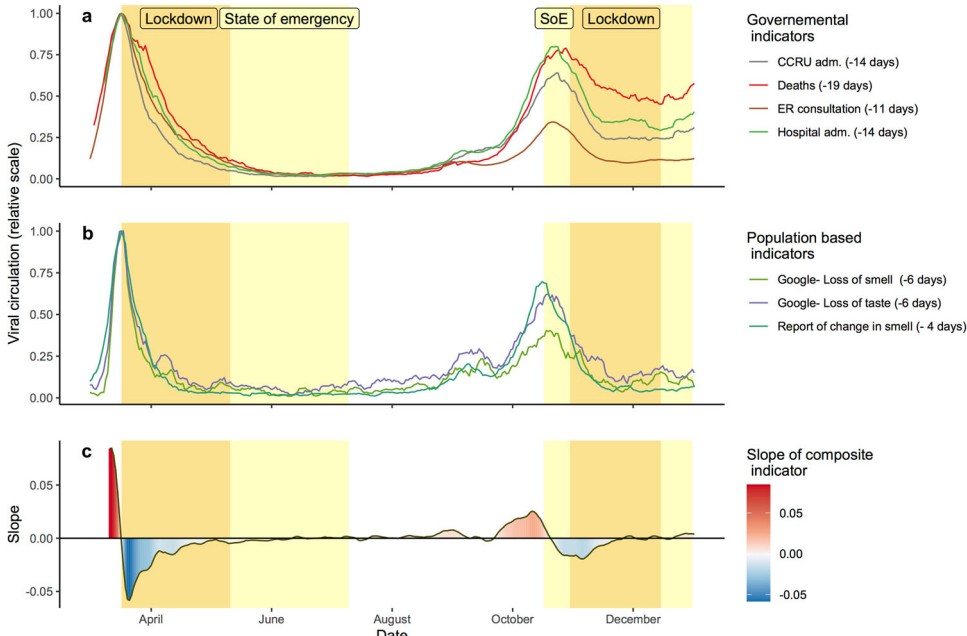

**Fig. 1 Estimation of the level of SARS-CoV-2 circulation in France in 2020.** All the trends have been shifted by their respective number of days to get the peaks to align for the first period of maximal virus circulation. **a** Phased governmental indicators: number of CCRU admissions, hospital admissions, number of daily deaths related to COVID, and ER ratio of consultations for suspected cases of COVID-19. **b** Population-based indicators: (1) level of online request for taste loss (perte gout), (2) smell loss (perte odorat), (3) day of first notice of smell change on the crowd-sourced survey. **c** Progression of the slope of the composite indicator (representing the mean of all indicators). *CCRU adm* critical care resuscitation unit admission, *ER* consultation emergency room consultation, *Hospital adm* hospital admission, *SoE* State of Emergency.

conservative approach since we grouped under curfew location where curfew was actually not implemented.

**Reporting summary**. Further information on research design is available in the Nature Research Reporting Summary linked to this article.

## Results

To monitor the circulation of the SARS-CoV-2, we collected surveillance indicator data from independent sources: (1) governmental statistics (https://geodes.santepubliquefrance.fr) healthcare system data compiled by the French health ministry: ER ratio of consultation for COVID-19, daily number of hospitalization for COVID-19, daily number of admission in CCRU for COVID-19, daily number of deaths for COVID-19, and (2) population-based tools that monitor chemosensory changes (smell and taste change linked to COVID-19 were followed by one crowd-sourced survey (https://project.crnl.fr/odorat-info/) and independently by following the search requests of French residents compiled by Google trends loss of taste (perte gout); loss of smell (perte odorat) https://trends.google.com/).

For each indicator, we computed the delay between the day of the highest value (date of the peak) and the maximum level of SARS-CoV-2 circulation in France (17th March[1,5–7]). The longest delay was 19 days for the indicator based on death caused by COVID-19. The shortest delay concerned chemosensory changes: 4 days for the first report of change in the sense of smell by survey participants, 6 days for the peak of online queries for smell and taste loss. The delay between the peak of viral circulation and the peak of COVID-19 diagnosis in the ER was 11 days and 14 days for admission to the CCRU. For governmental indicators, their granularity (availability at a local level) allows computing the delays for each statistic over all French departments, and based on

the distribution of delay compute a standard deviation and get an estimation of the variability. We obtained a standard deviation of 5 days for ER ratio, 6 days for CCRU admission and hospitalization, the largest variation is for the death SD = 8 days. The computation of the standard errors reveals values under 1 day for all indicators (ranging from 0.5 to 0.8) demonstrating a relatively low level of uncertainty. These results are consistent with the chronology of pathological symptomatology[4] (see discussion). Based on these numbers, we, therefore, phased all the indicators using a unique reference time point, namely the first peak in viral circulation (Fig. 1a and b). Furthermore, we computed the slope of a composite indicator based on the mean value of all the indicators (Fig. 1c).

The results show that a second peak in viral circulation in France was around the 21st of October based on the composite indicator. Five of the seven indicators converge to a peak within a 5-day period between Monday, 19th October and Saturday, 24th October. The crowd-sourced survey suggests a peak as early as October 16th. Contrary to the other indicators, the indicator based on death count shows a plateau -instead of a peak - with a maximal incidence the October 28th.

Besides the indicator based on death, most of the indicators suggest that the viral circulation reaches a stable low level 2 weeks after the beginning of the lockdown. An increase is observable on the governmental indicators between the 25th December and 1st January 2021 in France, which coincides with the Christmas and holiday season and the release of the lockdown in December. It is notable that all indicators converge to a maximal SARS-CoV-2 viral circulation during the state of emergency. Therefore, the decrease started before the lockdown (October 31st). An analysis of the slope of the composite indicator shows that the increase and decrease in viral circulation were less than during the first peak in the spring. Notably, the slope became negative after 21 October, suggesting that in France, the level of virus circulation

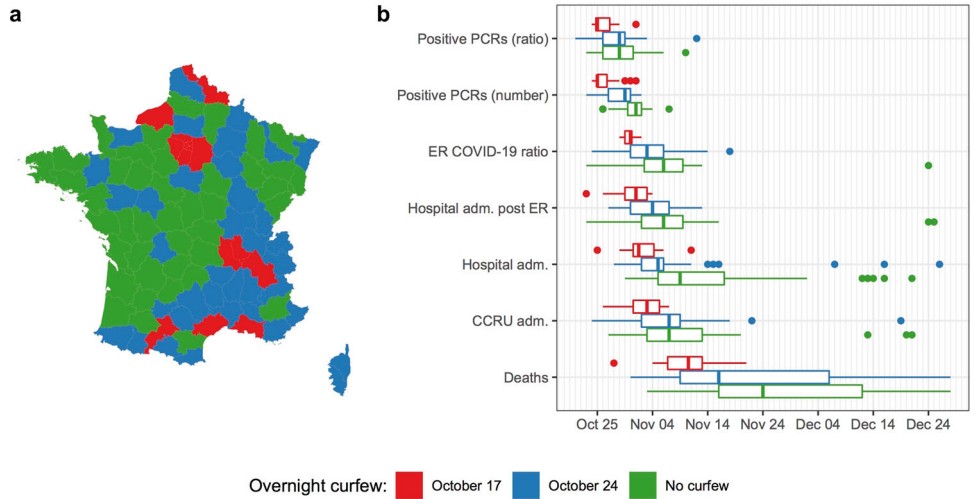

**Fig. 2 Impact of the overnight curfew. French departments were assigned a red ($n = 16$), blue ($n = 37$), or green ($n = 43$) label reflecting the date of implementation of overnight curfew enforcement. a** Geographic positions of the departments. **b** Date of the maximum value (peak) of seven epidemic indicators used to monitor SARS-CoV-2 circulation (unphased) in the departments according to the date of overnight curfew implementation. Hospital admission post ER represents the number of admissions to the hospital after ER consultation. Box plots follow standard Tukey representations. The lower and upper hinges correspond to the 1st and 3rd quartiles. The upper whisker extends from the hinge to the largest value no further than 1.5 * IQR from the hinge (where IQR is the inter-quartile range between the 1st and 3rd quartiles). The lower whisker extends from the hinge to the smallest value at most 1.5 * IQR of the hinge. Data beyond the end of the whiskers (outlying points) are plotted individually. *positive PCRs* positive Polymerase Chain Reaction, *ER* emergency room, *adm* admission, *CCRU adm* critical care resuscitation unit admission. *Map:* © OpenStreetMap contributors, open database license www.openstreetmap.org/copyright.

begins to decrease during the public health state of emergency before the autumnal lockdown.

It is important to note that temporally the state of emergency was not implemented homogeneously across all French territories. An overnight curfew was enforced at the start of the state of emergency (October 17th) in eight zones (including the entire Paris region). Later on, a curfew was implemented on October 24th in 38 other zones (Fig. 2). For governmental data, it is possible to estimate at the local scale the level of SARS-CoV-2 circulation in France in 2020 (Supplementary Figure 2). We computed the dates with the highest value (peak) for six epidemic indicators available at the local level (96 departments). All the indicators suggest that viral circulation decreased earlier in each location where curfew was implemented. All the indicators show significant differences between locations with an overnight curfew implemented at the beginning of the state of emergency and locations without curfew before the lockdown (t test, $p$ value $<1 \times 10^{-2}$). The peak of viral circulation in zones with a curfew implemented during the middle of the state of emergency shows an intermediate situation (in blue, Fig. 2). One indicator, the number of COVID-19 cases (i.e., the number of positive PCRs), showed the compelling result that zones under curfew reached a maximum value on October 25th, which is 5 days before the lockdown (October 31st), which confirms a decrease in viral circulation in these locations without a lockdown. As a control, we checked whether a similar difference between the same areas could be observed during the peak of viral circulation, which happened during spring 2020. For all $t$ test but one, $p$ value are non-significant and over 0.3. The only significant difference observed is a one single day difference between the peaks of one indicator (hospitalization after ER between) of areas with no curfew (green) and early curfew area (red). This difference is limited and might be due to an artifact considering that 1 day corresponds to the granularity of our data (numbers are reported day to day). Globally, the absence of differences between the areas, which is consistent with the fact that the spring lockdown

was applied homogeneously across the location (Supplementary Figure 3).

## Discussion

Overall, these results suggest that in France, the 2020 autumnal state of emergency and curfews were the initial triggers of the decrease of the viral circulation (Supplementary Data 1). However, state of emergency measures was not sufficient to maintain suppression once the subsequent lockdown was released for the holiday's season, suggesting the need for additional measures to limit the viral circulation.

These results are in keeping with previous results which suggest that night bars, full-service restaurants, as well as music-related events and gymnasiums play an important role in viral circulation[8,9]. It is important to note that these results were obtained using governmental statistics as well as population-based statistics collected from two non-governmental sources.

These results are partially based on the time phasing method. One first key of this method is the determination of the day of the highest level of virus transmission in spring. Modeling studies[6,7,10,11] pointed the 17th of March as the day with the highest level of virus transmission in France since the number of contacts and transmission was abruptly stopped by the very stringent lockdown in France. The lockdown was implemented at noon on the 17th, one possibility is that it might be slightly earlier than the 16th (rush in supermarkets and train stations) or even 14–15th (sunny weekend of the election before). It has to be noted that considering our conclusion of an early peak of viral circulation, the choice of the 17th is a conservative choice. Indeed, if we consider that the first peak was the 14th March, then it would mean that our indicators for the autumnal wave should have peak even earlier thereby still before the lockdown enforcement.

The second key of our method; the subtraction of the delay between the infection and specific indicators (such as first

symptoms, or death) to infer SARS-CoV-2 circulation history; is not original, see, for example, He et al.[12,13]. But one specificity of our approach, computing the delay at a population level is only possible due to the extraordinary events which have happened during spring 2020 in France: (1) the rapid spread of a new virus and (2) the highly stringent lockdown drastically decreasing the virus circulation and allowing to observe sharp peak and estimating such delay at a national scale level. This population-based approach is particularly adapted to the comparison between two periods of heavy SARS-CoV-2 circulation, separated by only a few months in the same population living in metropolitan France. With such a within-population comparison, the difference in terms of delay should therefore be limited. Moreover, the estimation of the delay is consistent with the chronology of the pathological symptomatology[4,5,14–17]. All indicators may not be addressed in literature but there are time estimates of the first symptoms of the illness onset to dyspnea, ICU admission, and death or discharge. The first symptoms, such as fever, appear on average 1 day after infection[4]. Taste and smell appear around 3 days after the first symptoms[5]. The delay from illness onset time to ARDS (acute respiratory distress syndrome) is 12 days, which might correspond to our estimation of the time of hospitalization. Depending on the study ICU admission range from 12 to 15 days. And finally, the time from illness onset to death ranges from 17 to 21 days depending on countries and studies[4,17]. Therefore, we observed consistency between these clinical observations and our own estimations based on population statistics.

However, we acknowledge that differences between the spring lockdown and the autumnal lockdown in terms of medical treatment, medical policy, and the availability of adapted materials in hospitals might have affected the temporal dynamics of some of the indicators that were analyzed. For example, we hypothesize that medical progress and the availability of new equipment, which have increased patient survival, could explain the plateau observed for the death-based indicator. However, all the indicators suggest a decrease in viral circulation before the lockdown. It is unlikely that medical progress has the same effect on all the indicators. Like health system indicators, present population-based indicators present also limitations. For example, the crowd-sourced survey might be impacted by a sampling bias and recall bias due to media coverage (see ref. [5] for discussion). For example, a participation bias toward urban regions with early curfew might explain the early maximum. Nevertheless, considering all phased indicators and the raw data (without phasing), the indicators based on viral nucleic-acid detection (positive ratio and raw number) confirm a decrease in viral circulation in France before the autumnal lockdown.

However, the decrease of the level of circulation of SARS-COV2 during the autumnal state of emergency is slower compared with the one observed in spring. Although the overall trend shows the positive impact of measures such as curfew the rate of change of these measures is also important. By being more restrictive, the lockdown will more efficiently stop the virus propagation and therefore might enable a more rapid decrease in cases and deaths.

The aim of the implementation of the lockdown in France being to reduce the pressure of overloading hospitals, one can ask whether the speed of reduction of viral circulation in France was fast enough during the state of emergency to reduce such pressure. The pressure can be measured by the number of patients in CCRUs. At its maximum, 9640 were occupied beds in France at the autumnal peak while it was 40% higher in spring (13,885). This number depends on the admission rate but also on the time of the outcome (amelioration of the patient state or death), which can vary. It is a reliable indicator of the hospital overload but not

of the virus circulation. When we applied the same rationale as our main analysis, we found a delay of 22 days between the peak of viral circulation level and the peak of CCRU occupancy level (Supplementary Figure 4). This analysis suggests that the state of emergency allowed to reach a plateau phase lower than in spring and a beginning of diminution of the hospital load, which continued during the lockdown period (Supplementary Figure 3). When case burden is high, even incremental changes in transmission can have significant effects on public health outcomes such as hospitalizations and deaths, and that while the state of emergency may have triggered a decline in transmission, the later lockdown might have been important to achieve the outcomes.

Our results show that an extended state of emergency could slow virus dissemination but it is logically less effective than a highly stringent lockdown (as implemented in March), which could reduce more dramatically the transmission rates down after reopening safely the country but at a high societal cost. The implementation of a lockdown requires important efforts on the part of inhabitants, even compared to a public health state of emergency associated with a curfew (the differences are detailed in Supplementary Data 1). One of the characteristics of the lockdown is the stay-at-home order forbidding non-essential movement outside individual homes. Except for essential activities (work, shopping for necessities, medical exams, etc.), inhabitants were limited to one outing per day for a maximum of one hour, within a radius of 1 km. A travel waiver certificate was mandatory in order to leave home which enabled police enforcement. Social activity such as collective sports or visits to friends and family were forbidden. On 31 October, stores such as bookstores, clothing stores, and full-service restaurants were closed and this had important socio-economic consequences. The economical and psychological long-term effects of the lockdown compared with curfew are yet to be known. Therefore, further studies are necessary to compare the efficiency/utility of short lockdowns vs longer state of emergency on the global population health, i.e., not only the virus circulation.

Although we show that the level of virus circulation decreased during the period of emergency of state and curfew. It is important to point out that we do not know the key factors of this decrease and that the same measure might not have the same impact on a population depending on the timing. For example, one could suggest that school holidays in October have impacted the decrease of the circulation level, but it is unlikely to be the main factor of decrease since holidays were homogeneously implemented in France at the difference of curfew. Also, prior experience with the pandemic and better preparedness may also have resulted in more appropriate changes in individual's behavior when cases were rising again. Therefore, it may be possible that less harsh restrictions are more effective in reducing transmission than they would have been early in the epidemic. One other explanation might be that the implementation of a state of emergency and curfews could also have indirect impacts on people's behavior in the virus circulation. The population may tend to relax health rules during long periods of a pandemic so the implementation of a state of emergency could be an efficient reminder that the virus is still circulating. Also, people who have already experienced the prior spring lethal pandemic period, could retake the safe sanitary practices and implement appropriate behavior when recalling that covid-19 cases are rising again. Inversely, the removal of lockdown limitations just before the period of family and friend celebration (Christmas and new year) might be an indirect signal of relaxation to the population despite the official continuation of the state of emergency. Therefore, following people's adherence to covid-safe practices (through surveys for example) before or in parallel to implementing harsher measures should be useful.

In conclusion, although the effect of a strict lockdown on viral circulation was demonstrated in spring in France, it was not the major factor that initiated the decrease in viral circulation in autumn 2020. The decrease was initiated during the public health state of emergency associated with the curfew. From a public health standpoint, it might be important to limit the cost of measures to control viral circulation as much as possible. Indeed, these measures have a direct (less exercise, postponement of medical treatment, mental health) and indirect (degradation of economic status) effect on individual health.

In contrast, an increase in virus circulation is observable at the release of the lockdown for Christmas and holiday's season, suggesting that the efficiency of specific measures is dependent on the country's dynamics (seasons, new variants etc.). Therefore, desirable public health outcomes might be obtained by balancing measures with different levels of restrictiveness.

At the time of writing this manuscript (27 April 2021), after several months of the state of emergency and curfew, we observed a new rise of COVID-19 cases, leading to a heavy load on CCRU in France. In consequence, a lockdown was enforced on 20th March 2021 for 16 departments including major cities: i.e., Paris and Nice. On the 3rd April, a lockdown was enforced on all other departments. Despite these measures as well as an increased vaccination rate (21% of the French population is now vaccinated with a first dose, 60% of individuals over 65 years old), the load in CCRU does not decrease strongly as observed during the previous lockdowns. This suggests that the effect of public health measures varies in time and depends on numerous factors.

## Data availability
Online queries were downloaded from Google Trends (https://trends.google.com/), using the R library gtrendsR. We searched separately for the popularity of the terms: loss of smell (perte odorat) and loss of taste (perte gout), using default selection of All categories and Web search, within the timeframe of 1 March 2020 to 10 January 2021. Data of the online survey conducted in French population between 8 April and 10 January 2021 are available in the supplementary Data 2 (sheet: survey_QUALITYofLIFE). The Governmental indicators regarding the healthcare system in France (ER consultation, hospitalizations, CCRU admission and deaths, PCR) were downloaded on 25 February 2021 and are available on https://www.data.gouv.fr/fr/pages/donnees-coronavirus and https://geodes.santepubliquefrance.fr. The authors confirm that all relevant data are included in the article in Supplementary Data 2.

## Code availability
Code for data cleaning and analysis associated with this manuscript is available at: https://github.com/DenisPierron/COVID-19/blob/main/codeCOM.zip.

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

## Acknowledgements
This work was financed by EXTREM-O/Confinez 2.0 (CNRS, CNES) and the COR-ODORAT grant (IDEX-Lyon—Université de Lyon). We are grateful to the anonymous reviewers for their comments that helped us to improve this communication.

## Author contributions
Study conception and design: D.P.; writing the paper: D.P., V.P.L., and M.B.; data acquisition and curation: D.P., V.P.L., M.B., C.F., C.B., A.M.P., and analysis: D.P., V.P.L., T.L., M.B., and C.B. writing, edition, and approval of the final manuscript: V.P.L., A.M.P., L.T., M.C.S., C.B., C.F., T.L., M.B., D.P.

## Competing interests
The authors declare no competing interests.
