## [Peer Review File · Communications Medicine]

Reviewers' comments:

Reviewer #1 (Remarks to the Author):

This manuscript uses the difference in time between the peak number of cases in lockdown one and a number of different indicators to predict when cases began to decline in the autumn wave. The authors find that all indicators suggested cases were in decline before the start of the second strict lockdown period and thus conclude that the second lockdown was not the major factor for initiating the decrease in cases. They then repeat this analysis at the department level, since curfew measures were not homogeneously applied across all regions, and find the regional indicators also suggest a decline in cases before lockdown.

Main comment:

1. This results in this paper heavily rely on the assumption that the delays between the different indicators did not change between the two different phases of the epidemic in France. The author partially addresses this in the discussion section but I think this section would benefit from strengthening. Some suggestions are as follows:
 - A. Do the delays between the peak cases and indicators agree with what is found in literature. I understand that not all indicators may not be addressed in literature but there are estimates between symptoms onset and death, for example.
 - B. If you repeat the analysis for the first wave at the regional level do you get the same delays for all the regions or are they distributed?
 - C. Do deaths continue to decline further into December since the data stops relatively soon after the decrease begins?
 - D. Are you able to add in some uncertainty to your estimates? The peak in the death indicator seems very close to the implementation of the second lockdown. Have you accounted for weekend reporting effects?
 - E. it would be useful to see the case count over time for readers who are not so familiar with the French data.
2. There is no mention in the text about the numbers of people in hospital at the time of the second lockdown. Despite cases reducing, the speed of reduction may not be fast enough with only a "state of emergency" to reduce the pressure of overloading hospitals that the authors mention in the introduction.

Minor comments:

1. In the manuscript the authors write "we were able to synchronize and timely phase all surveillance indicators with the actual level of virus circulation." Is this a new method you propose using? or is it widely done in literature?
2. It would be useful to have a list of all your indicators in the text somewhere. Do you weight all these indicators equally or are some more reliable than others?
3. Please cite your data sources where you first mention them in the text or let the reader know where to look for more information about them in the data section.
4. I'm a little bit confused by what Figure 1 is showing. For other countries deaths have continued to increase into the lockdown period to reflect the people who were infected just before we went into lockdown. Have the trends been shifted by their respective number of days (e.g. 19 for deaths) to get the peaks to align for lockdown 1? Please make this clearer in the caption.
5. Please state in the text what methods the 17th of March was chosen as the highest level of virus transmission - data? modelling studies? Or is this the day France went into lockdown?

Reviewer #2 (Remarks to the Author):

See attached.

In the current manuscript, authors use temporal data on several direct and indirectly associated Covid-19 factors compiled from governmental data such as hospital admissions & deaths as well as from population based indicators such as google trends (queries) about loss of smell & taste and from survey to understand the role of different types of population-level mitigation approaches taken by French government for controlling covid-19 transmission. According to their findings, data suggests that viral circulation had already beginning to decline in France before the harsh lockdowns were implemented in autumn. Given the negative economic and mental impact of lockdowns on individuals and countries as a whole, it is important to assess it's utility and to explore if viral circulation can be curtailed with relatively less restrictions. Current manuscript contributes towards this understanding, However, I do not believe that the manuscript conclusively shows that lockdowns are not necessary at all. I have a couple of comments that I hope authors would find constructive and will incorporate in their narrative.

1. While the data suggests that viral circulation declines during the state of emergency before lockdown was implemented in France, there are other factors that are important to understand whether it was possible to avoid lockdown. One of them being geographical heterogeneity of the spread. State of emergency may have been able to slow down the spread but may not have stopped it from widespread disease throughout the country. Including subnational trends of the data to see if similar patterns that were observed nationally were observed subnationally too may shed some light on this.
2. Another important comparison between the trends of virus circulation is the rate at which decline is occurring. Comparing the sharp declines during the first lockdown with the decline observed during the state of emergency period towards the end of year reveals that rate of decline in circulation of virus is much slower. While the slower decline makes sense, it may not have been able to sufficiently reduce the circulation quick enough. Whereas following lockdown (even if shorter than what was implemented) could potentially bring the transmission rates down at a much faster rate allowing the country to reopen safely. A discussion about utility of short lockdowns vs longer state of emergency would be useful in the manuscript.
3. Prior experience with the pandemic and better preparedness may also have resulted in more appropriate changes in individual's behaviours when cases were rising again. Therefore, it may be possible that less harsh restrictions is more effective in reducing transmission than it would have been early in the epidemic. Therefore, a discussion about taking people's adherence to covid-safe practices (through surveys) into account before implementing lockdown would be useful.

Thanks to the two reviewers for their accurate comments and constructive suggestions. We think that the incorporation of them to the manuscript has resulted in a considerable improvement of the clarity and readability of the paper.

Reviewer #1 Remarks:

“This manuscript uses the difference in time between the peak number of cases in lockdown one and a number of different indicators to predict when cases began to decline in the autumn wave. The authors find that all indicators suggested cases were in decline before the start of the second strict lockdown period and thus conclude that the second lockdown was not the major factor for initiating the decrease in cases. They then repeat this analysis at the department level, since curfew measures were not homogeneously applied across all regions, and find the regional indicators also suggest a decline in cases before lockdown”

Main comment:

1. This results in this paper heavily rely on the assumption that the delays between the different indicators did not change between the two different phases of the epidemic in France. The author partially addresses this in the discussion section but I think this section would benefit from strengthening. Some suggestions are as follows:

A. Do the delays between the peak cases and indicators agree with what is found in literature. I understand that not all indicators may not be addressed in literature but there are estimates between symptoms onset and death, for example.

> We thank the reviewer for suggesting to compare our estimation with data available in the literature. We were able to identify several studies reporting the time from the illness onset to the different stages of hospital care.

>DISCUSSION: Moreover, the estimation of the delay *are consistent with the chronology of the pathological symptomatology* ^{4,5,14-17}. *All indicators may not be addressed in literature but there are time estimates of the first symptoms of the illness onset to dyspnoea, ICU admission, and death or discharge. The first symptoms, such as fever, appear on average 1 days after infection⁴. Taste and smell appear around 3 days after the first symptoms⁵. The delay from illness onset the time to ARDS (Acute respiratory distress syndrome) is 12 days which might correspond to our estimation of time of hospitalization. Depending on the study ICU admission range from 12 to 15 days. And finally the time from illness onset to death ranges from 17 to 21 days depending on countries and studies ^{4,17}. Therefore we observe a consistency between these clinical observations and our own estimations based on population statistic.*

B. If you repeat the analysis for the first wave at the regional level do you get the same delays for all the regions or are they distributed?

> We thank the reviewer for suggesting this analysis. In the revised version, as suggested we added the study of the delay at the local level.

> **RESULTS:** *As a control, we checked whether a similar difference between the same areas could be observed during the peak of viral circulation which happened during spring 2020. For all t-Test but one, p-value are non-significant and over 0.3. The only significant difference observed is a one single day difference between the peaks of one indicator (Hospitalization after E.R between) of areas with “no curfew” (green) and early curfew area (red). This difference is limited and might be due to an artefact considering that 1 day corresponds to the granularity of our data (numbers are reported day to day). Globally the absence of differences between the areas which is consistent with the fact that the spring lockdown was applied homogeneously across the location (Supplementary figure 3).*

Supplementary figure 3 : Date of the maximum value (peak) of 7 epidemic indicators used to monitor SARS-CoV-2 circulation (unphased) in the departments during the spring lockdown 2020 according to the date of overnight curfew implementation of autumn 2020. Adm.=Admission. Hospital admission postE.R represent the number of admissions to hospital after ER consultation.

C. Do deaths continue to decline further into December since the data stops relatively soon after the decrease begins?

> In the revised version, we added the data corresponding to the whole month of December and added a discussion on it.

> **RESULTS :** Beside the indicator based on death, most of the indicators suggest that the viral circulation reaches a stable low level two week after the beginning of the lockdown. An increase is observable on the governmental indicators between the 25 December and 1st January (Christmas holidays) in France.

> **Figure 1 :**

D.1 Are you able to add in some uncertainty to your estimates? The peak in the death indicator seems very close to the implementation of the second lockdown.

> We thank the reviewer for suggesting this point. Following the first remark, we were able to compute the standard error for each governmental indicator based on the study at the local level. In concordance with bibliography data (discussed earlier), the indicator based on death rate is the one with highest variability, and thus it might be the less reliable to estimate the virus circulation at a daily/weekly scale (in contrast it is considered as reliable indicator of the global pandemic impact in a long-term period). In the discussion we also discuss the fact that the shortage of resources during the first wave might have influenced the rapidity of the occurrence of death in some departments. Another factor being that the better knowledge and equipment might have differed the date of death between the first and second wave.

> **RESULTS :** *For governmental indicators, their granularity (availability at a local level) allows to compute the delays for each statistic over all French départements and based on the distribution of delay compute a standard deviation and get an estimation of the variability. We obtained a standard deviation of 5 days for ER ratio, 6 days for CCRU admission and hospitalization, the largest variation is for the death $SD= 8$ days. The computation of the standard errors reveals values under 1 day for all indicators (ranging from 0.5 to 0.8) demonstrating a relatively low level of uncertainty.*

D.2 Have you accounted for weekend reporting effects?

> We presented this point in the methods, we use a standard method by computing a rolling average over 7 days.

> **METHODS** : (Statistics section): In order to account for weekend reporting effects and random fluctuation, we computed a rolling average over 7 days, for each indicator.

E. it would be useful to see the case count over time for readers who are not so familiar with the French data.

> We added this point in introduction with a new figure in the supplementary data.

> **INTRODUCTION** : The case of the French population provides a useful framework to address this issue, indeed in 2020 two periods of heavy *SARS-CoV-2* circulation have been reported based on the number of deaths and cases (Supplementary Figure 1).

Supplementary figure 1: ECDC indicators of the COVID-19 pandemic in France during the year 2020. Periods of lockdown are pictured in orange and periods of emergency state in yellow.

2. There is no mention in the text about the numbers of people in hospital at the time of the second lockdown. Despite cases reducing, the speed of reduction may not be fast enough with only a “state of emergency” to reduce the pressure of overloading hospitals that the authors mention in the introduction.

> We thank the reviewer for this comment and added this point in the discussion.

DISCUSSION: *The aim of the implementation of the lockdown in France being to reduce the pressure of overloading hospitals, one can ask whether the speed of reduction viral circulation in France was fast enough during “state of emergency” to reduce such pressure. The pressure can be measured by the number of patients in critical care resuscitation units. At its maximum, 9,640 were occupied beds in France in the autumnal peak while it was 40% higher in spring (13,885). This number depends on the admission rate but also on the time of the outcome (amelioration of the patient state or death) which can vary. It is a reliable indicator of the hospital overload but not of the virus circulation. When we applied the same rationale as our main analysis, we found a delay of 22 days between the peak of viral circulation level and the peak of CCRU occupancy level (figure S3). This analysis suggests that the “state of emergency” allowed to reach a plateau phase lower than in spring and a beginning of decrease of the hospital load which continued during the lockdown period (figure S3).*

Supplementary figure 4: Day to day number of COVID-19 patients in Critical Care Units (load) in France during the year 2020. Orange rectangles represent periods of lockdown and yellow rectangle represent period of sanitary emergency state. The grey line represents the real-time value reported by the French health system. The blue line represents the same data shifted by 22 days following our strategy to phased the peak with the peak of maximum of virus circulation the 17th march.

Minor comments:

1. In the manuscript the authors write “we were able to synchronize and timely phase all surveillance indicators with the actual level of virus circulation.” Is this a new method you propose using? or is it widely done in literature?

> We developed this point in the discussion.

DISCUSSION : A second key of our method; the subtraction of the delay between the infection and specific indicators (such as first symptoms, or death) to infer *SARS-CoV-2* circulation history; is not original, see for example He et al. 2020^{12,13}. But one specificity of our approach is computing the delay between infection and indicators at a population level (similar to the recent publication of Lu et Reis 2021¹⁴). This approach is only possible due to the “extraordinary” events which have happened during spring 2020 in France: (1) the rapid spread of a new virus and (2) the highly stringent lockdown drastically decreasing the virus circulation and allowing to observe sharp peak and estimating such delay at a national scale level. This population-based

approach is particularly adapted to the comparison between two periods of heavy SARS-CoV-2 circulation, separated by only a few months in the same population living in metropolitan France.

2. It would be useful to have a list of all your indicators in the text somewhere. Do you weigh all these indicators equally or are some more reliable than others?

> We clarified this point in the methods and results.

>**METHODS** : *Then we phased all indicators (i.e 1- google loss of smell ; 2- google loss of taste ; 3- report of change in smell; “ER consultation” , “hospitalization”, “CCRU admission” , “COVID deaths”) to represent virus circulation on specific days (figure 1). We also define a composite indicator by computing the mean of all indicators and compute its rolling slope on 7 days. All indicators were weighted equally.*

> **RESULTS**: *To monitor the circulation of the SARS-CoV-2, we collected surveillance indicator data from independent sources: (1) governmental statistics (<https://geodes.santepubliquefrance.fr> healthcare system data compiled by the French health ministry: ER ratio of consultation for COVID-19 , daily number of hospitalization for COVID-19, daily number of admission in CCRU for COVID-19, daily number of deaths for COVID-19 and (2) population-based tools that monitor chemosensory changes (smell and taste change linked to COVID-19 were followed by one crowdsourced survey (<https://project.crn.fr/odorat-info/>) and independently by following the search requests of French residents compiled by Google trends “perte gout” ; “perte odorat”; "<https://trends.google.com/>").*

3. Please cite your data sources where you first mention them in the text or let the reader know where to look for more information about them in the data section.

> We agree and we have clarified this point in the result part (confere the previous modification (minor point number 2)).

4. I'm a little bit confused by what Figure 1 is showing. For other countries deaths have continued to increase into the lockdown period to reflect the people who were infected just before we went into lockdown. Have the trends been shifted by their respective number of days (e.g. 19 for deaths) to get the peaks to align for lockdown 1? Please make this clearer in the caption.

> We clarified this point. Indeed, the trends have been shifted by their respective number of days to get the peaks to align for lockdown 1.

Caption figure 1 : *Estimation of the level of SARS-CoV-2 circulation in France in 2020. All the trends have been shifted by their respective number of days to get the peaks to align for the first period of maximal virus circulation.*

5. Please state in the text what methods the 17th of March was chosen as the highest level of virus transmission - data? modelling studies? Or is this the day France went into lockdown?

> We clarified this point.

DISCUSSION : *These results are partially based on the time phasing method. One first key of this method is the determination of the day of the highest level of virus transmission in spring. Modelling studies^{6,7,10,11} pointed the 17th of March as the day with the highest level of virus transmission in France since the number of contacts and transmission was abruptly stopped by the very stringent lockdown in France. The lockdown was implemented at noon the 17th, One possibility is that it might be slightly earlier than the 16th (rush in supermarkets and train stations) or even 14-15th (sunny week-end of election before). It has to be noted that considering our conclusion of an early peak of viral circulation, the choice of the 17th is a conservative choice. Indeed, if we consider that the first peak was the 14th March, then it would mean that our indicators for the autumnal wave should have peak even earlier thereby still before the lockdown enforcement.*

Reviewer 2 Remarks:

1. « While the data suggests that viral circulation declines during the state of emergency before lockdown was implemented in France, there are other factors that are important to understand whether it was possible to avoid lockdown. One of them being geographical heterogeneity of the spread. State of emergency may have been able to slow down the spread but may not have stopped it from widespread disease throughout the country. Including subnational trends of the data to see if similar patterns that were observed nationally were observed subnationally too may shed some light on this. »

> *We agree with the reviewer that we need to clarify the subnational trends of the data compared to the national data so we have included in the supplemental figures to address this point.*

> **RESULT:** It is important to note that temporally the state of emergency was not implemented homogeneously across all French territories. Overnight curfew was enforced at the start of the state of emergency (October 17th) in 8 zones (including the entire Paris region). Later on, a curfew was implemented on October 24th in 38 other zones (Figure 2). For governmental data it is possible to estimate at the local scale of the level of SARS-CoV-2 circulation in France in 2020 (Supplementary figure 2). We computed the dates with the highest value (peak) for 6 epidemic indicators available at the local level (96 departments). All the indicators suggest that viral circulation decreased earlier in each location where curfew was implemented.

Supplementary figure 2. Départemental variation of the estimation of the level of SARS-CoV-2 circulation in France in 2020. Each line represents one département. All the trends have been shifted by their respective number of days to get the peaks to align for the first peak. For clarity only département representing more than 1% of the COVID-19 hospitalization have been presented. Color of the lines represents the categorization of the département during the autumnal peak. A) Phased governmental indicators: number of CCRU admissions, Hospital admissions, Number of daily deaths related to COVID and ER ratio of consultations for suspected cases of COVID-19

2. « Another important comparison between the trends of virus circulation is the rate at which decline is occurring. Comparing the sharp declines during the first lockdown with the decline observed during the state of emergency period towards the end of year reveals that rate of decline in circulation of virus is much slower. While the slower decline makes sense, it may not have been able to sufficiently reduce the circulation quick enough. Whereas following lockdown (even if shorter than what was implemented) could potentially bring the transmission rates down at a much faster rate allowing the country to reopen safely. A discussion about utility of short lockdowns vs longer state of emergency would be useful in the manuscript »

> We thank the reviewer for raising this point. In the revised version, we added a paragraph to complement our initial discussion.

DISCUSSION : “However, the decrease of level of circulation of SARS-COV2 during the autumnal state of emergency is slower compared to the one observed in spring. Therefore, our results show that an extended state of emergency could slow virus dissemination but it is logically less effective than a highly stringent lockdown which could reduce more dramatically the transmission rates down after reopening safely the country but at a high societal cost. Economical and psychological long-term effect of the lockdown compared to curfew are yet to be known. Further studies are necessary compare the efficiency/utility of short lockdowns vs longer state of emergency on the global

population health (and not only the virus circulation). The aim of the implementation of the lockdown in France being to reduce the pressure of overloading hospitals, one can ask whether the speed of reduction viral circulation in France was fast enough during “state of emergency” to reduce such pressure. The pressure can be measured by the number of patients in critical care resuscitation units. At its peak, 9,640 were occupied beds in France in the autumnal peak while it was 40% higher in spring (13,885). This number depends on the admission rate but also on the time of the outcome (amelioration of the patient state or death) which can vary. It is a reliable indicator of the hospital overload but not of the virus circulation. When we applied the same rationale as our main analysis, we found a delay of 22 days between the peak of viral circulation level and the peak of CCRU occupancy level (figure S3). This analysis suggests that the “state of emergency” allowed to reach a plateau phase lower than in spring and a beginning of diminution of the hospital load which continued during the lockdown period (figure S3) ”

3. Prior experience with the pandemic and better preparedness may also have resulted in more appropriate changes in individual’s behaviours when cases were rising again. Therefore, it may be possible that less harsh restrictions is more effective in reducing transmission than it would have been early in the epidemic. Therefore, a discussion about taking people’s adherence to covid-safe practices (through surveys) into account before implementing lockdown would be useful.

> We agree with the reviewer. We modified the new version to better discuss the indirect effect of people’s better adherence to covid-safe practices by the prior experience with the pandemic in the spring first wave.

DISCUSSION : *While we show that the level of virus circulation decreased during the period of emergency of state and curfew. It is important to point out that we do not know the key factors of this decrease and that the same measure might not have the same impact on a population depending on the timing. For example, one could suggest that school holidays in October have impacted the decrease of the circulation level, but it is unlikely to be the main factor of decrease since holidays were homogeneously implemented in France at the difference of curfew. Also, prior experience with the pandemic and better preparedness may also have resulted in more appropriate changes in individual’s behaviours when cases were rising again. Therefore, it may be possible that less harsh restrictions is more effective in reducing transmission than it would have been early in the epidemic. One other explanation might be that the implementation of a state of emergency and curfews could also have indirect impacts on people behavior the virus circulation. Population may tend to relax health rules during long periods of a pandemic so the implementation of state of emergency could be an efficient reminder that the virus is still circulating. Also people who have already experienced the prior spring lethal pandemic period, could retake the safe sanitary practices and implement appropriate behavior when recalling that covid-19 cases are rising again. Inversely, the removal of lockdown limitations just before the period of family and friend celebration (Christmas and new year) might be an indirect signal of relaxation to the population despite the official continuation of state of emergency. Therefore, following people’s adherence to covid-safe practices (through surveys for example) before or in parallel to implementing harsher measure should be useful.*

REVIEWERS' COMMENTS:

Reviewer #1 (Remarks to the Author):

The authors have responded to my comments well and I am happy for this paper to proceed.

Reviewer #2 (Remarks to the Author):

I am happy with the revisions. Thank you for this important work.